# Intraocular dendritic cells characterize HLA-B27-associated acute anterior uveitis

**Maren Kasper[1†], Michael Heming[2†], David Schafflick[2], Xiaolin Li[2], Tobias Lautwein[2], Melissa Meyer zu Horste[3], Dirk Bauer[1], Karoline Walscheid[1,4], Heinz Wiendl[2], Karin Loser[5], Arnd Heiligenhaus[1,6]*, Gerd Meyer zu Hörste[2]***

[1]Ophtha-Lab, Department of Ophthalmology, and Uveitis Centre at St. Franziskus Hospital, Münster, Germany; [2]Department of Neurology with Institute of Translational Neurology, University Hospital Muenster, Muenster, Germany; [3]Augen-Zentrum-Nordwest Augenpraxis Ahaus, Ahaus, Germany; [4]Department of Ophthalmology, University of Duisburg-Essen, Essen, Germany; [5]Department of Human Medicine, University of Oldenburg, Oldenburg, Germany; [6]University of Duisburg-Essen, Essen, Germany

**Abstract** Uveitis describes a heterogeneous group of inflammatory eye diseases characterized by infiltration of leukocytes into the uveal tissues. Uveitis associated with the HLA haplotype B27 (HLA-B27) is a common subtype of uveitis and a prototypical ocular immune-mediated disease. Local immune mechanisms driving human uveitis are poorly characterized mainly due to the limited available biomaterial and subsequent technical limitations. Here, we provide the first high-resolution characterization of intraocular leukocytes in HLA-B27-positive (n = 4) and -negative (n = 2) anterior uveitis and an infectious endophthalmitis control (n = 1) by combining single-cell RNA-sequencing with flow cytometry and protein analysis. Ocular cell infiltrates consisted primarily of lymphocytes in both subtypes of uveitis and of myeloid cells in infectious endophthalmitis. HLA-B27-positive uveitis exclusively featured a plasmacytoid and classical dendritic cell (cDC) infiltrate. Moreover, cDCs were central in predicted local cell-cell communication. This suggests a unique pattern of ocular leukocyte infiltration in HLA-B27-positive uveitis with relevance to DCs.

**\*For correspondence:**
arnd.heiligenhaus@uveitis-zentrum.de (AH);
gerd.mzh@uni-muenster.de (GMzH)

†These authors contributed equally to this work

**Competing interest:** The authors declare that no competing interests exist.

## Introduction

Uveitis describes a heterogeneous group of inflammatory diseases involving uveal tissues in the intraocular cavity of the eye. Non-infectious uveitis is regarded as an autoimmune disorder and is associated with various immune-mediated systemic diseases. According to the Standardization of Uveitis Nomenclature (SUN) working group, uveitis is classified based on the anatomical location of uveitis as anterior, posterior, intermediate, and panuveitis (*Jabs et al., 2005*). Anterior uveitis (AU) primarily affects the iris and ciliary body and constitutes its most frequent type (approximately 80 % of cases) resulting in permanent vision loss through, for example, secondary cataract, glaucoma, or macular edema (*Rothova et al., 1996*; *Thorne et al., 2016*).

Acute anterior uveitis (AAU) associated with the HLA haplotype B27 (B27+ AAU) is the most common and often severe form of uveitis. Its typical clinical features include acute onset of discomfort, eye redness, tearing, visual impairment, and excessive cellular infiltration in the aqueous humor (AqH; ie, the intraocular liquid of the anterior chamber) that is devoid of immune cells under non-diseased conditions. The prevalence of the HLA-B27 haplotype is approximately 8–10% among Caucasians and 40–70% among AAU patients (*Kopplin et al., 2016*) and thus represents a strong genetic risk factor for AAU (*Brewerton et al., 1973*; *Huhtinen and Karma, 2000*). The HLA-B27 allele also conveys an increased genetic risk for other immune-mediated diseases, including spondyloarthropathies (SpA)

**eLife digest** Uveitis is a form of inflammation in the eye. It can occur in response to infection, or when the immune system mistakenly attacks the eye, in what is known as autoimmune uveitis. In approximately 80 percent of cases, the front part of the eye is affected. During an inflammatory episode, the liquid inside the front part of the eye fills with immune cells, but the nature of these cells remains unknown. This is because uveitis is rare, and doctors cannot routinely take samples from inside the eyes of affected individuals to diagnose the disease. This lack of samples makes research into this disease challenging.

There are two main groups of immune cells that could be responsible for uveitis: myeloid cells and lymphoid cells. Myeloid cells form the first line of immune defense against infection by non-specifically attacking and removing pathogens . Lymphoid cells form the second line of immune defense, attacking specific pathogens. Lymphoid cells also have long-term memory, meaning they can 'remember' previous infections and fight them more effectively. Lymphoid cells receive instructions from a type of myeloid cell called a dendritic cell about what to attack. Dendritic cells relay their instructions to lymphoid cells using molecules called human leukocyte antigens (HLA). Autoimmune uveitis affecting the front part of the eye is common in individuals with an HLA type called HLA-B27, suggesting that communication between dendritic and lymphoid cells plays an important role in this type of inflammation.

To make the most of limited patient samples, Kasper et al. used single cell techniques to examine the immune cells from the fluid inside the eye. Six samples came from people with autoimmune uveitis, and one from a person with an eye infection. The infection sample contained mainly myeloid cells that might attack bacteria responsible for the infection. In contrast, the autoimmune uveitis samples contained mainly lymphoid cells. Of these samples, four were from individuals with the gene that codes for the HLA-B27 molecule. These samples had a unique pattern of immune cells, with more dendritic cells than the samples from individuals that did not have this gene.

This study included only a small number of individuals, but it shows that analysing single immune cells from the eye is possible in uveitis. This snapshot could help researchers understand the local immune response in the eye, and find an optimal treatment.

and inflammatory bowel disease (IBD). B27+ AAU is thus regarded as a prototypic ocular immune-mediated disease and, accordingly, its treatment is based on corticosteroids (*Khan et al., 2015*; *Rosenbaum, 2015*) and classical and biological disease-modifying anti-rheumatic drugs (*Heiligenhaus et al., 2012*; *Bou et al., 2015*). Notably, although T-cell inhibition (eg, by cyclosporine A) is beneficial in other uveitis entities (*Nussenblatt et al., 1991*), it provides limited efficacy in B27+ AAU (*Gómez-Gómez et al., 2017*), suggesting a yet poorly defined role of innate immune cells in its pathogenesis.

Immune cells infiltrating the anterior chamber of the eye can be observed clinically, but are difficult to *obtain* for further analyses. In fact, the invasive sampling of AqH is rarely clinically justified. AqH fine-needle aspirates are primarily applied for the verification of, for example, infectious uveitis (*Chronopoulos et al., 2016*) and rarely to unravel the pathogenesis of non-infectious uveitis (*Greiner and Amer, 2008*; *Chen et al., 2015*; *de Groot-Mijnes et al., 2015*; *Zhao et al., 2015*). Our knowledge of AqH-infiltrating leukocytes and underlying mechanisms of uveitis thus remains superficial, despite its frequency and severity.

In AAU, a role of bacterial triggers has been proposed (*Huhtinen et al., 2002a*; *Huhtinen et al., 2002b*). Therefore, innate immune cells such as monocytes and dendritic cells (DCs) that phagocytose and process foreign antigens are of special interest. In fact, phenotyping of circulating monocytes in patients with immune-mediated uveitis emphasized differences during the disease course (*Liu et al., 2015*; *Walscheid et al., 2016*; *Walscheid et al., 2018*; *Kasper et al., 2018*). The microbiome, intestinal barrier dysfunction, and immune response have also been suggested to contribute to the pathogenesis of AAU (*Rosenbaum and Asquith, 2018*). Soluble mediators (*Bauer et al., 2020*; *Abu El-Asrar et al., 2020*; *Bauer et al., 2018*; *Bonacini et al., 2020*) and infiltrating leukocytes have been analyzed (*Denniston et al., 2012*) previously in the AqH of uveitis patients, but access to samples and technical challenges remain the main bottlenecks toward better understanding the pathomechanisms in uveitis.

Single-cell RNA-sequencing (scRNA-seq) studies have provided unprecedented high-resolution insights into immune mechanisms in various tissues (*Heming et al., 2021*; *Wolbert et al., 2020*). Therefore, we here combined scRNA-seq with flow cytometry and protein analysis of soluble cytokines and thereby provide the first partially unbiased characterization of leukocytes in the AqH from patients with B27+ AAU compared to patients with HLA-B27-negative AU (B27-AU) and acute infectious endophthalmitis. We found that lymphocytes predominate in the intraocular infiltrate in B27+ AAU and specifically showed an elevated frequency of plasmacytoid DCs (pDCs) and classical DCs (cDCs). In B27+ AAU, DCs featured the most predicted intercellular interactions and increased expression of AAU- and SPA-related genome-wide association study (GWAS) risk genes that distinguished this uveitis from active B27-AU. This suggests a specific involvement of DCs in B27+ AAU, and subtypes of AU thus exhibit specific patterns of local leukocyte responses.

## Results

### Single-cell analysis of AqH requires optimized sample processing and recruitment

We here aimed for an unbiased characterization of leukocytes in the AqH in AU flares. We screened 4980 total patients with any intraocular inflammation seen at our uveitis center. We included 11 patients with current onset of uveitis flare (n = 8) or endophthalmitis (n = 3) into this study, being untreated with topical corticosteroids for this flare (*Supplementary file 1a*), corresponding to a recruitment rate of approximately 0.18 %. In 10 of the AqH samples, cytokine analysis was performed. Cellular infiltrates were analyzed via scRNA-Seq in AqH samples of four patients with active B27+ AAU, two patients with active B27-AU, and one patient with active bacterial endophthalmitis (*Streptococcus pneumoniae*); five of those samples were analyzed in parallel via flow cytometry (*Supplementary file 1a*). Deep characterization of AqH-infiltrating leukocytes is thus feasible.

The endophthalmitis patients were significantly older than B27+ AAU patients (analysis of variance (ANOVA), p = 0.0157; *Supplementary file 1a*). There were four females in the B27-AU cohort, and three males and one female in the B27+ AAU cohort. All B27+ AAU patients had associated SpA, and two of them received adalimumab therapy. Both uveitis groups did not differ significantly regarding age (ANOVA, p = 0.566), Antinuclear antibodies (ANA) status, frequency of systemic anti-inflammatory treatment, topical medication, previous ocular surgery, or time since uveitis onset.

### Single-cell transcriptomics reconstructs key leukocyte lineages in aqueous humor

We then performed scRNA-seq of ocular infiltrating cells from fresh AqH fine-needle aspirates (*Supplementary file 1a*). We defined 5000 input cells as our maximum intended input per sample and used excess cells beyond that number for flow cytometry analysis. Thereby, we obtained transcriptional information of 13,550 total individual cells and 1936 average cells (± 1411 SD) per patient with 830 average genes (± 402 SD) detected per cell (*Supplementary file 1b*). This is in accordance with the expected cell recovery rate of 50 % of the scRNA-seq technique we employed (*Zheng et al., 2017*). After quality control, we clustered the single-cell (sc) transcriptomes of all patients combined and identified 13 individual cell clusters (*Figure 1A and B*). We manually annotated these clusters based on the expression of marker genes (*Figure 1C*, *Figure 1—figure supplement 1*, *Supplementary file 1c*). As previously shown in AqH aspirates of human uveitis patients (*Denniston et al., 2012*), only hematopoietic cell clusters were identified. Clusters were broadly classified into cells of myeloid (40 %; cDCa, cDCb, pDC, mature DC (matDC), granulo, myeloid) and of lymphoid origin (60 %; natural killer (NK), γδ T cells (gdTC), CD8, regulatory T (Treg), CD4, naïve B cells (Bc), plasma). Myeloid clusters separated into cDC clusters tentatively named cDCa (*ITGAX, CLEC7A*) and cDCb (*CLEC10A, MRC1*), pDC (*IL3RA*/CD123, *CLEC4C*/CD303) and matDC (*TMEM176B, IDO1, FSCN1, LAMP3, CD83*), granulocytes (granulo; S100A12/ *S100 A8 high, CCL2 low*), and myeloid cells with unclear assignment (myeloid; S100A12/*S100 A8 low, CCL2 high*) (*Figure 1C*). The marker genes expressed by the cDC sub-clusters did not fully overlap with previously described cDC type 1/2 markers used to identify DC subsets from cerebrospinal fluid (DC1: *CLEC9A, XCR1, BATF3*; DC2: *CD1C, FCER1A, CLEC10A*) (*Heming et al., 2021*) or peripheral blood (DC1: *CLEC9A, C1ORF54, HLA-DPA1, CADM1, CAMK2D*;

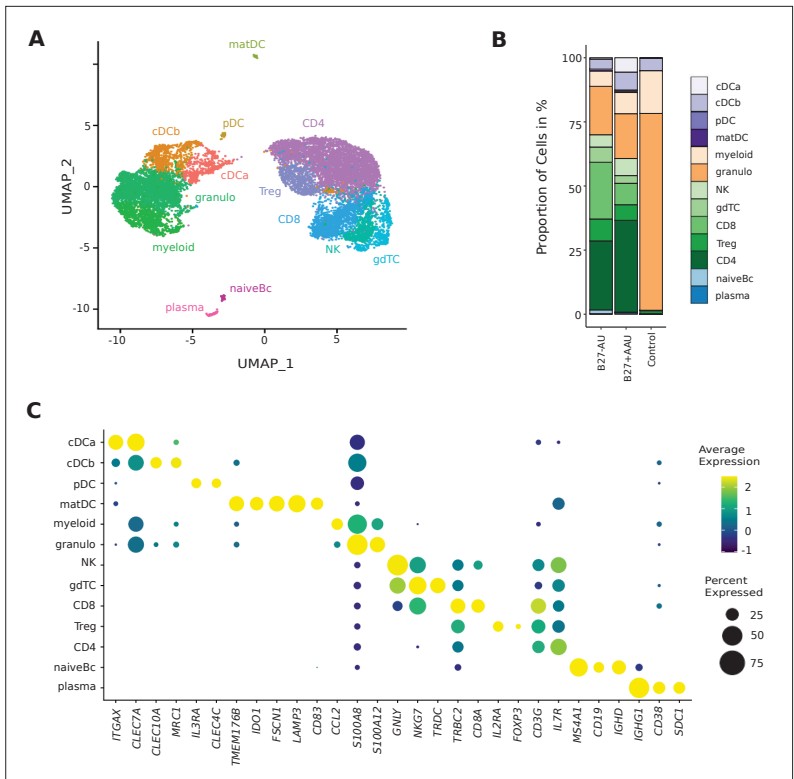

**Figure 1.** Single-cell transcriptomics reconstructs leukocyte subsets infiltrating the eye. (**A**) Uniform manifold approximation and projection (UMAP) projection of seven pooled samples (control n = 1; B27-AU n = 2; B27+ AAU n = 4). The single-cell (sc) transcriptomes were manually annotated to cell types based on marker gene expression and distinguished in 13 cell clusters (color-coded; each dot represents one cell). (**B**) The mean proportion of cells (%) in each cluster per group is depicted in a stacked bar plot. (**C**) Dot plot of selected marker genes grouped by cluster. The average gene expression level is color-coded, and the circle size represents the percentage of cells expressing the gene. Threshold was set to a minimum of 10 % of cells in the cluster expressing the gene. DC: dendritic cell, pDC: plasmacytoid DC, matDC: mature DC; granulo: granulocytes, NK cells: natural killer cells, gdTC: γ δ T cells, Treg cells: regulatory T cells, Bc: B cells.

The online version of this article includes the following figure supplement(s) for figure 1:

**Figure supplement 1.** Feature plots of lineage marker genes.

DC2: CD1C, FCER1A, CLEC10A, ADAM8) (***Villani et al., 2017***). We therefore intentionally named these clusters cDCa/b to prevent ambiguity.

Lymphoid clusters were classified as CD4+ T cells (CD4; *IL7R, CD3G*), CD8+ T cells (CD8; *CD8A, CD3G*), gdTC (*NKG7, TRDC*), Treg cells (*IL2RA, FOXP3*), NK cells (*GNLY, NKG7*), naive Bc (*MS4A1, CD19, IGHD*), and plasma cells (plasma; *IGHG1, CD38, SDC1*/CD138) (***Figure 1C***). We thus identified all major leukocyte lineages in AqH fine-needle aspirates (***Figure 1B***, ***Supplementary file 1d***).

## Diverse uveitis entities exhibit a unique intraocular leukocyte composition

We next sought to understand how intraocular inflammation differed between uveitis entities. First, we assessed the endophthalmitis control patient and found almost exclusively myeloid lineage clusters with predominating granulocytes in accordance with an acute anti-bacterial response (***Figure 2B***; ***Figure 2—figure supplement 1***; ***Engstrom et al., 1991***).

In contrast, the cellular infiltrates in B27-AU and B27+ AAU patients' AqH were of more lymphoid origin, reflecting their autoimmune etiology (***Figure 2A and B***). When systematically comparing B27+ AAU vs B27-AU samples, inter-patient variability was high (***Figure 2A and B***), but several populations still substantially differed between uveitis subtypes (***Figure 2C***, ***Supplementary file 1d***). While the

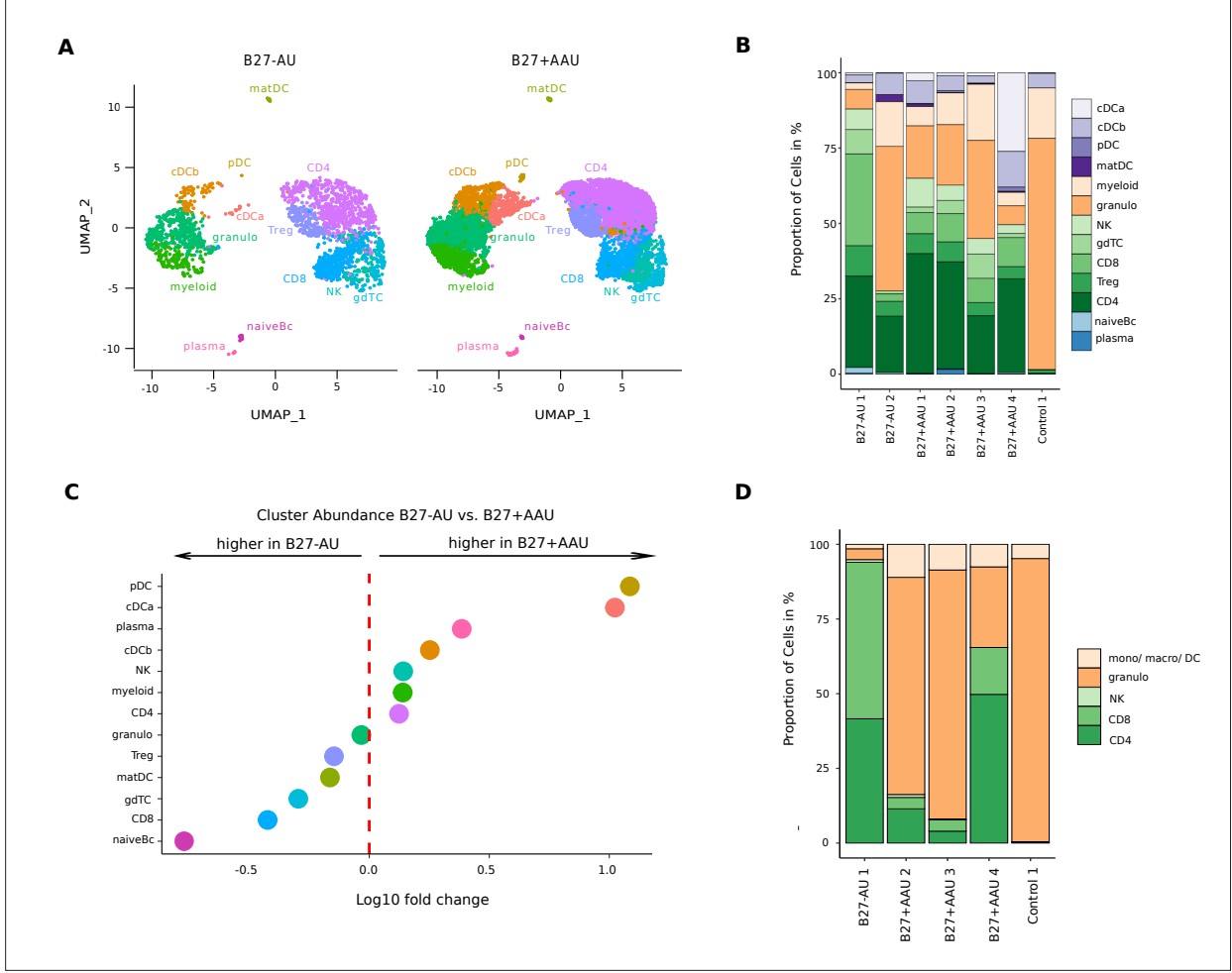

**Figure 2.** A unique intraocular leukocyte pattern characterizes individual uveitis causes. (**A**) UMAP projection of pooled B27-AU (n = 2) versus pooled B27+ AAU (n = 4) samples. The single-cell (sc) transcriptomes were manually annotated to cell types based on marker gene expression and distinguished in 13 cell clusters (color-coded; each dot represents one cell). (**B**) The proportion of cells (%) in each cluster is depicted in a stacked bar plot for individual samples. (**C**) Dot plot of cluster abundance of B27-AU versus B27+ AAU. The x axis represents the decadic logarithm of fold change of proportional cluster size. (**D**) Box plots of proportion of cells (%) of cDCa and pDC from B27-AU and B27+ AAU. The boxes show the median, and the lower and upper quartile. Whiskers include 1.5 times the interquartile range of the box. The overlaid dots represent individual observations. (**E**) Leukocytes of aqueous humor (AqH) samples were analyzed according to their frequency (%) of granulocytes, monocytes/macrophages/DCs, CD4+ and CD8+ T cells, and NK cells. The proportion of each cell population identified via flow cytometry is depicted in a stacked bar plot. DC: dendritic cell, pDC: plasmacytoid DC, matDC: mature DC; granulo: granulocytes, NK cells: natural killer cells, gdTC: $\gamma\delta$ T cells, Treg cells: regulatory T cells, Bc: B cells, mono: monocyte, macro: macrophage.

The online version of this article includes the following figure supplement(s) for figure 2:

**Figure supplement 1.** Individualized scRNA-seq results of anterior chamber-derived leukocytes.

**Figure supplement 2.** Flow cytometric analysis of aqueous humor samples.

CD8 and naive Bc clusters were reduced, the pDC and cDCa clusters were more abundant in B27+ AAU patients (*Figure 2C and D*), overall indicating an influx or expansion of DC in this uveitis entity.

Next, we sought to confirm our findings using flow cytometry (*Figure 2E*; *Figure 2—figure supplement 2*). The expression of canonical lineage markers differs between mRNA and protein quantification (*Peterson et al., 2017*), and we therefore used widely applicable pan-lineages in flow cytometry. AqH-derived cells of some of the patients (*Supplementary file 1a*) were analyzed to distinguish granulocytes (CD3-CD11b+HLA-DR-), monocytes/macrophages/DCs (CD3-CD11b+HLA-DR+), CD4+ and CD8+ T cells (CD3+CD11b-), and NK cells (CD3-CD11b-CD56+). This confirmed the mainly myeloid infiltrate in the endophthalmitis patient (*Figure 2E*). B27-AU patients showed an infiltrate dominated by T cells, low NK cells, and low cells of myeloid origin. In B27+ AAU patients, myeloid cells were more abundant than

in B27-AU (*Figure 2E*). Also, the abundance of broad cell classes quantified by scRNA-seq and flow cytometry showed a positive correlation (*Figure 2E*, *Figure 2—figure supplement 2B*). The higher frequency of granulocytes detected in flow cytometry than in the scRNA-seq might be due to the higher fragility of these cells during processing for scRNA-seq (*Zilionis et al., 2019*). Overall, subtypes of uveitis were thus characterized by a unique pattern of local inflammatory cells.

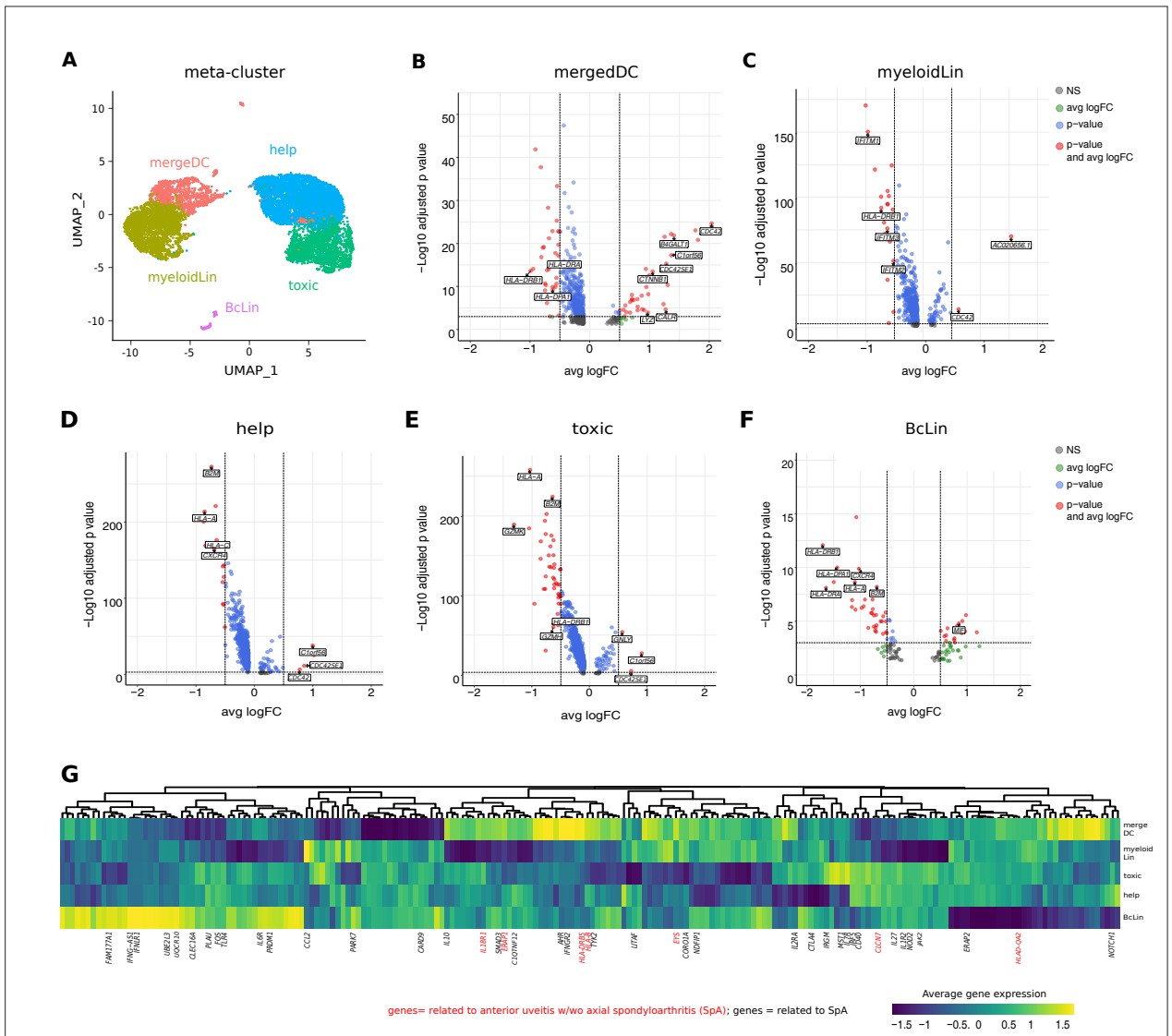

**Figure 3.** Intraocular leukocytes express a subtype-specific transcriptional phenotype. (**A**) UMAP projection of pooled B27-AU (n = 2) versus pooled B27+ AAU (n = 4) samples. The single-cell (sc) transcriptomes were manually annotated to cell types based on marker gene expression and distinguished in five meta-clusters (color-coded; each dot represents one cell). Differentially expressed (DE) genes of B27-AU vs B27+ AAU of each meta-cluster (**B**) *mergeDC* (matDC, pDC, DCa, cDCb), (**C**) *myeloidLin* (myeloid, granulo), (**D**) *help* (Treg, CD4), (**E**) *toxic* (CD8, NK, gdTC), (**F**) and *BcLin* (naïve Bc, plasma) are depicted as volcano plots. The threshold for average natural logarithmic fold change (avg logFC) was set to 0.5 and for adjusted p-value, to 0.001. Selected genes are labeled. (**G**) Heatmap showing differences in average gene expression (B27+ AAU minus B27-AU) of genome-wide association study (GWAS) risk genes (*Supplementary file 1g*). Data were scaled column wise. Columns were clustered using euclidean distance measure and complete linkage. Yellow color indicates a higher expression in B27+ AAU samples, and blue color indicates a higher expression in B27-AU samples. Risk genes for anterior uveitis and for spondyloarthropathies (SpA) are labeled in red and black, respectively. DC: dendritic cell, pDC: plasmacytoid DC, matDC: mature DC; granulo: granulocytes, NK cells: natural killer cells, gdTC: γ δ T cells, Treg cells: regulatory T cells, Bc: B cells, *BcLin*, B-cell lineage.

## Specific transcriptional phenotype of intraocular leukocytes in subtypes of uveitis

Next, we sought to understand how local leukocytes differ transcriptionally between uveitis entities. To analytically account for low total cell numbers, we merged clusters into five broad cell type of 'meta-clusters' for differential expression analysis (*Figure 3A*, *Supplementary file 1e*): helper T cells (*help*; Treg, CD4), cytotoxic cells (*toxic*; CD8, NK, gdTC), merged DCs (*mergeDC*; matDC, pDC, cDCa, cDCb), other myeloid cells (*myeloidLin*; myeloid, granulo), and B-cell lineage (*BcLin*: naive Bc, plasma). We then tested for differentially expressed (DE) genes between B27+ AAU and B27-AU. Across all clusters, multiple major histocompatibility complex (MHC) class I and class II related genes (*HLA-A*, *HLA-DPA1*, *HLA-DRA*, *HLA-DRB1*, *B2M*) were expressed at lower levels in B27+ AAU samples (*Figure 3B–F*). Furthermore, in B27+ AAU, the *help* and *BcLin* meta-clusters downregulated one cytokine receptor (*CXCR4*). The *toxic* meta-cluster downregulated signs of cytotoxicity (*GZMK*, *GZMH*, *LTA*). The *BcLin* meta-cluster featured an increase of macrophage-inhibitory factor (*MIF*), known to inhibit NK cell activity (*Apte et al., 1998*). The *myeloid* meta-cluster reduced expression of several interferon (IFN)-regulated genes (*IFITM* genes; *Figure 3B–F*).

Elevated *CTNN1* expression in mergeDC (*Figure 3B*) pointed to an involvement of the WNT/catenin pathway as previously described for SpA (*Xie et al., 2016*). Increased expression of *Lyz* and its cognate antisense *AC020656.1,* and *CALR* (*Liu et al., 2016*) and its antisense *AC092069.1,* within the mergeDC cluster suggested an activation of myeloid cells and simultaneously local counter regulating mechanisms. Elevated expression of *B4GALT1* in mergeDC, in detail cDCa and cDCb (*Supplementary file 1f*), potentially reflected the migratory capacity of DC (*Johnson and Shur, 1999*). Notably, the mergeDC, help, and toxic meta-clusters all upregulated *C1ORF56*, an oncogene previously found induced in activated lymphocytes in IBD (*Uniken Venema et al., 2019*) and splenic NK cells (*Crinier et al., 2018*).

We also found elevated expression of prefoldin 5 (*PFDN5*) in *myeloid*, *toxic,* and *help* clusters (*Supplementary file 1e*), which was recently described as a specific marker for B27+ AAU associated with SpA (*Kwon et al., 2019*). Furthermore, as previously shown in joint biopsies of SpA patients (*Carlberg et al., 2019*; *Lam et al., 2019*) and in association with autoinflammatory diseases (*Carlberg et al., 2019*; *Lam et al., 2019*), elevated expression of *CDC42* was detected in mergeDC, myeloid, and help clusters and of *CDC42SE1* in mergeDC, myeloid, help, and toxic clusters (*Figure 3B–E*).

The risk for developing HLA-B27-associated autoimmune diseases is partially determined by variants in non-HLA genetic loci (*Robinson et al., 2015*; *Lin et al., 2011*; *Australo-Anglo-American Spondyloarthritis Consortium (TASC) et al., 2010*; *Li et al., 2019*; *Evans et al., 2011*; *International Genetics of Ankylosing Spondylitis Consortium (IGAS) et al., 2013*; *Trochet et al., 2019*; *Ellinghaus et al., 2016*; *Huang et al., 2020*). We therefore interlinked our transcriptional data with existing genetic information, by testing which 'meta-clusters' differentially expressed AAU/SpA risk genes between B27+ AAU and B27-AU (*Figure 3G*, *Supplementary file 1g*). Both uveitis-related and SpA-related risk genes were included in the analysis because all B27+ AAU patients had systemic SpA (*Supplementary file 1*). Many SpA-related risk genes were highly expressed in the *BcLin* cluster (eg, *CLEC16A*, *IFNLR1*), and in the mergeDC cluster (eg, *IFNGR2*, *AHR*) in B27+ AAU (*Figure 3G*, *Supplementary file 1g*). Compared to B27-AU, expression of genes involved in the detection of pathogens (eg, *CARD9*, *TLR4*) showed lower expression in the mergeDC cluster in B27+ AAU. Notably, most AU-related risk genes were preferentially expressed in the B27+ AAU mergeDC cluster (eg, *EYS*, *HLA-DRB5*, *ERAP1*, and *IL18R1*). This indicates the relevance of DC in B27+ AAU.

## Subtype-specific local leukocyte communication in uveitis

We also attempted to understand how uveitis controlled the local inter-cellular signaling circuitry. We therefore used a computational tool (CellPhoneDB; *Efremova et al., 2020*) to predict cell-cell interactions between human leukocytes in uveitis from scRNA-seq data. The pDC and plasma clusters had to be excluded because of their small size in B27-AU. In the resulting analysis, the granulo and myeloid clusters had the highest number of predicted interactions (*Figure 4—figure supplement 1*). In both uveitis groups, most interactions were between myeloid, granulo, gdTC, and multiple DC clusters. Myeloid lineage clusters thus express the highest capacity for cell-cell interaction.

Next, we tested for differences of predicted interactions between both subtypes of uveitis. We calculated the number of predicted interactions in B27+ AAU minus that in B27-AU (*Figure 4A*).

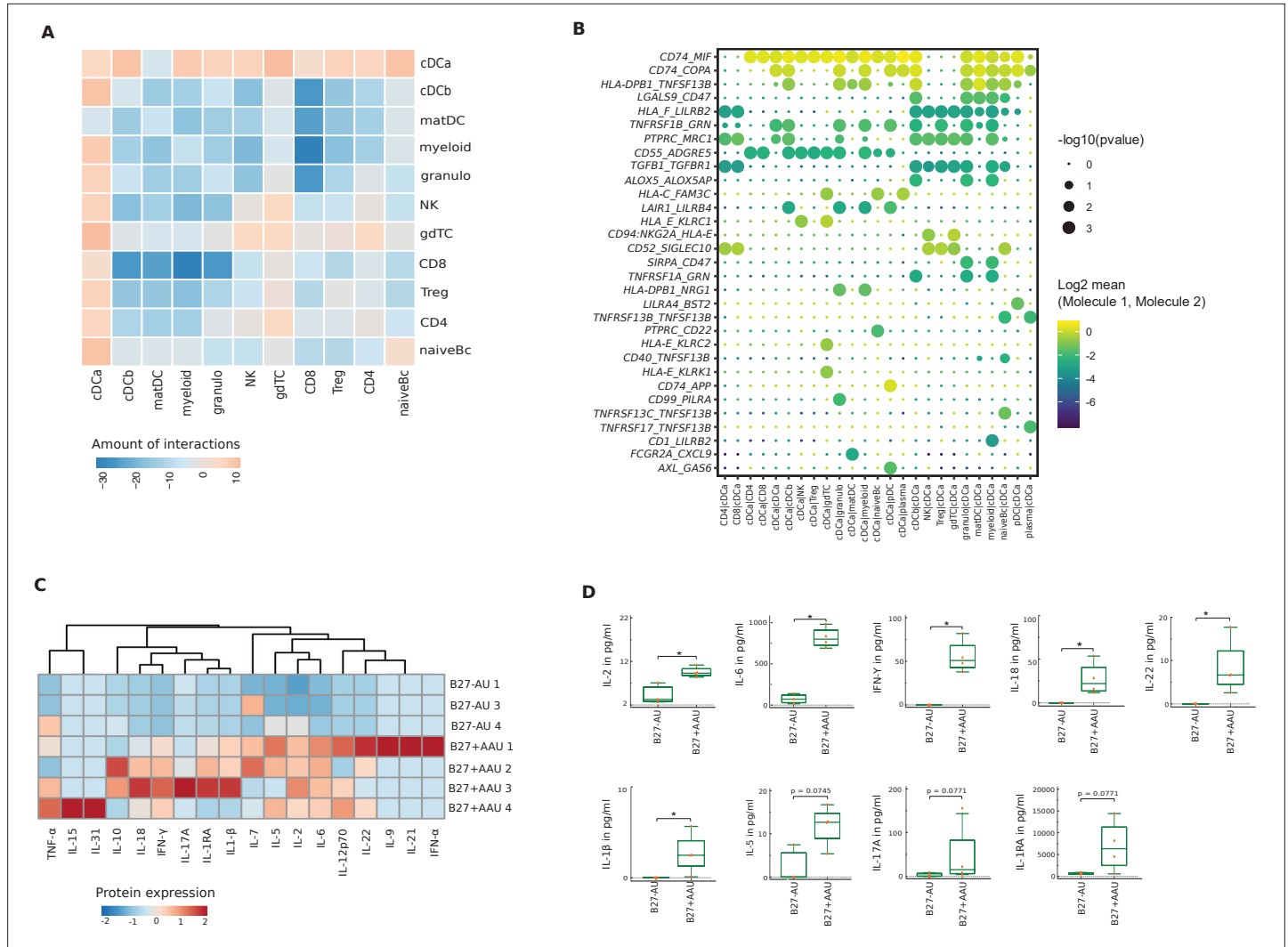

**Figure 4.** Altered local inter-cellular signaling in uveitis. (**A**) The total count of receptor-ligand interactions between cell clusters of B27-AU and B27+ AAU single-cell (sc) transcriptomes was obtained with CellPhoneDBv2.0 (see *Figure 4—figure supplement 1* for separate analyses). The heatmap shows the differences between B27+ AAU and B27-AU (amount of predicted interactions of all cell types, excluding plasma and plasmacytoid DC (pDC) due to few cells (<10) of B27+ AAU minus those of B27-AU). (**B**) Overview of all ligand–receptor interactions of the cDCa cluster with at least one significant interaction. Circle size indicates the p-values. The means of the average expression level of interacting molecule 1 in cluster 1 and interacting molecule 2 in cluster 2 are color-coded. (**C**) Heatmap showing aqueous humor (AqH) cytokine level of B27-AU (n = 3) and B27+ AAU (n = 4) patients (see *Supplementary file 1h* showing single values). Data were scaled column wise. Columns were clustered using euclidean distance measure and complete linkage. (**D**) Box plots of interleukin (IL)-2, IL-6, interferon (IFN)-$\gamma$, IL-18, IL-22, IL-1$\beta$, IL-5, IL-17A, and IL-1 receptor antagonist (IL-1RA) (pg/ml) in the AqH of patients with B27-AU and B27+ AAU. Dots represent individual data. Mann-Whitney U-test (*p<0.05).

The online version of this article includes the following figure supplement(s) for figure 4:

**Figure supplement 1.** Detailed results of the interactome prediction analysis.

**Figure supplement 2.** Cytokine expression in meta-cluster and cytokine serum level.

B27-AU displayed many interactions between T/NK clusters with myeloid, matDC, and cDCb clusters (*Figure 4A*, *Figure 4—figure supplement 1A*). In contrast, the cDCa cluster showed widespread interactions in B27+ AAU (*Figure 4A*, *Figure 4—figure supplement 1B*.). This suggests that preferential interactions differ between uveitis entities and that the cDCa cluster might be involved in intraocular inflammation in B27+ AAU.

Notably, when focusing on individual interactions of the cDCa cluster (*Figure 4B*), MHC class II-related transcripts (eg, *CD74, HLA-E*) were predicted to interact with surface molecules such as *MIF* or *KLRC1* with immunomodulatory capacity and known association to autoimmunity (*Figueiredo*

*et al., 2018*; *Borrego et al., 1998*). The predicted interaction of cDCa with pDC clusters in B27+ AAU samples also included the *LAIR-LILRB4* and *AXL-GAS6* interaction pairs (*Figure 4B*) that exert immunosuppressive phagocytosing functions (*Scutera et al., 2009*; *Brown et al., 2009*). Overall, our findings indicate subtype-specific local inter-cellular leukocyte signaling and indicate that DC forms a central signaling node in uveitis.

## Increased level of pro-inflammatory cytokines in AAU patients

Next, we sought to characterize the intraocular immune response through the analysis of soluble mediators. We therefore quantified a predefined set of cytokines in the AqH of these patient groups (*Supplementary file 1h*). Several cytokines were below detection limits, and the most notable feature of all these samples was a pronounced presence of interleukin (IL)-6 and IL-1 receptor antagonist (IL-1RA). High levels of immunosuppressive IL1-RA in AqH of uveitis/endophthalmitis patients have also been shown in B27+ AAU patients (*Zhao et al., 2015*; *de Vos et al., 1994*; *Planck et al., 2012*) and may reflect the immune-privileged microenvironment in the anterior chamber (*Zhao et al., 2015*; *de Vos et al., 1994*; *Planck et al., 2012*; *Dana et al., 1998*). The cytokine milieu in endophthalmitis was characterized by innate-related IL-6, tumor necrosis factor (TNF)-α, and IL-1β (*Supplementary file 1h*), which are involved in ocular barrier breakdown and leukocyte recruitment into ocular tissue (*Hao et al., 2016*; *Feys et al., 1994*; *Giese et al., 1998*).

Across uveitis entities, the interindividual heterogeneity of cytokine patterns was high (*Figure 4C and D*). When comparing uveitis entities, the B27+ AAU group showed significantly increased levels of IL-2, IL-6, IL-18, IL-22, IL-1β, and interferon (IFN)-γ (*Figure 4C and D*).

When comparing this with the scRNA-seq dataset, cytokine expression was identified across all 'meta-clusters' (*Figure 4—figure supplement 2A*), but not all cytokines were detected (eg, *IL31*, *IL5*, *IL9*, *IL21*, and *IFNA1*). Single-cell transcriptomics is generally prone to false negatives (*Vieth et al., 2019*), and stromal cells and resident immune cells in the iris can also express proinflammatory cytokines (*Miyamoto et al., 1999*), which may explain the discrepancy between techniques.

We additionally performed analysis of B27+ AAU and B27-AU serum samples, recruited during the active uveitis disease stage in the context of another project (*Kasper et al., 2018*; *Kasper, 2020*). There, we found significantly elevated IL-1RA and IFN-γ levels in the serum of B27+ AAU as compared to B27-AU patients (*Figure 4—figure supplement 2B,C*; *Supplementary file 1i*). This indicates a potential role of IFN-γ in both intraocular and systemic immune response during uveitis activity in patients with SpA.

## Discussion

Here, we provide the first high-resolution characterization of rare and difficult-to-obtain intraocular leukocytes in AU. By making scRNA-seq per se applicable to the AqH, we found that intraocular leukocytes are transcriptionally and compositionally different not only from blood (*Zheng et al., 2017*) but also from other tissues affected by autoimmune diseases such as synovial fluid in rheumatoid arthritis (*Stephenson et al., 2018*) or cerebrospinal fluid in multiple sclerosis (*Schafflick et al., 2020*). By comparing HLA-B27-positive versus -negative AU, we identified a unique composition of intraocular leukocytes in B27+ AAU. Notably, B27+ AAU showed an elevated frequency of pDc and cDC. In B27+ AAU, cDC had the highest amounts of predicted intercellular interactions and increased expression of AAU- and SpA-related GWAS risk genes that distinguished this uveitis from active B27-AU. This indicates the relevance of cDC in promoting intraocular inflammation.

Our findings allow diverse speculations on the pathogenesis of HLA-B27-associated AAU. We found T cells more activated in B27-AU than in B27+ AAU in accordance with the poor response of B27+ AAU to T-cell inhibition (*Gómez-Gómez et al., 2017*). The cell-to-cell network analysis revealed an involvement of *TNFSF13B* (*BAFF*), connecting naive Bc and plasma cells to cDCa. A previous study has shown enhanced levels of the B-cell factors BAFF and APRIL in AqH of AAU patients during inactive uveitis, pointing to a role of intraocular B cells in AAU (*Wildschütz et al., 2019*). Notably, naive Bc expressing CLEC16A (*Rijvers et al., 2020*), and plasma cells, expressing several SpA risk genes, were preferentially enriched in B27+ AAU. Also, plasma cells and the intraocular plasma cluster do not express *MS4A1*/CD20 and, accordingly, SpA patients (with or without AAU) mostly do not benefit from treatment with the anti-CD20 antibody rituximab (*Wendling et al., 2012*).

Several lines of evidence—including our study—indicate that SpA-associated B27+ AAU may be driven by innate immune cells triggered through innate pattern recognition of endogenous or exogenous pathogens. One study identified CD14+ monocytes/macrophages and a separate CD14−/low BDCA-1+ (CD1c) myeloid DC population in AqH of patients with AU (*Denniston et al., 2012*). This lends support to a central hub function of DC.

DCs—as professional antigen-presenting cells—orchestrate the interplay between innate and adaptive immunity. The classification of DCs is complex. Traditionally, three DC subsets were defined as classical (or myeloid) DC type 1 (cDC1) or type 2 (cDC2), and pDC (*Merad et al., 2013*). Additional subdivisions and the definition of monocyte-derived DCs further complexify the field (eg, *Villani et al., 2017*; *Yin et al., 2017*; *Collin and Bigley, 2018*). We identified four intraocular DC clusters in two different uveitis entities. As the marker genes identified in our study partially differed from cDC1/cDC2 in other tissues (*Heming et al., 2021*; *Villani et al., 2017*), we used a distinct annotation as cDCa/b. We also identified a novel DC cluster that expressed genes related to activation and maturity (*FSCN1, CD83, LAMP3*), and was therefore designated as matDC cluster. LAMP3 was previously shown to mediate migration from tumor to regional lymph nodes and exert regulatory capacity on other cells (*Zhang et al., 2019*). Furthermore, we identified DC5 cells, characterized by their high expression of *AXL* and *SIGLEC6* (*Villani et al., 2017*), within the cDCa, matDC, and pDC cluster. Expression of immune regulatory *MIF* and *AXL* in cells of cDCa cluster and their multiple predicted interactions with other infiltrating immune cells point to an immune regulatory and/or phagocytosing role of these cells during AAU.

Pathogens are one potential trigger of autoimmune uveitis in mice and intestinal and joint inflammation in rats overexpressing human HLA-B27 antigen (*Taurog et al., 1994*). Anti-*Saccharomyces cerevisiae* antibodies are elevated in patients with SpA and point to an involvement of CARD9 or NOD2 (*Maillet et al., 2016*). Transcriptional candidate risk genes related to innate immunity (eg, *CARD9, NOD2, TLR4*) were expressed at a lower level in the DC populations of B27+ AAU patients, questioning the role of bacterial and fungal pathogens in intraocular immune response.

In scRNA-seq, the alarmins S100A8/S100A9 are assigned mainly as marker genes for granulocytes and other phagocytes (*Zheng et al., 2017*). Furthermore, elevated S100A8/A9 expression points to an involvement of toll-like receptor (TLR) signaling and activation of innate immune cells. Thus, elevated S100A8/A9 and S100A12 levels serve as activity biomarkers of diverse autoimmune diseases (*Pruenster et al., 2016*; *Baillet et al., 2010*; *Vogl et al., 2018*; *Walscheid et al., 2015*). In particular, S100A8/A9 serum levels are increased constitutively in patients with HLA-B27-associated uveitis (*Kasper, 2020*; *Wang et al., 2016*). In scRNA-seq, the alarmins S100A8/S100A9 are assigned mainly as marker genes for granulocytes and other phagocytes (*Zheng et al., 2017*). However, expression levels of S100A8/A9 did not differ between the myeloid lineage clusters of B27-AU and B27+ AAU samples.

The analysis of the cytokine pattern of AqH and sera in the study confirmed previous observations describing elevated IL-18 and IFN-γ levels in the AqH (*Zhao et al., 2015*; *Lacomba et al., 2000*), and an elevated serum IFN-γ level in B27+ AAU patients during active uveitis (*Chen et al., 2015*; *Lacomba et al., 2000*). In particular, IL-18 is a main costimulatory factor for IFN-γ expression in T and NK cells (*Okamura et al., 1995*). Further, the elevated expression of several IFN-γ-related transcripts in B27+ AAU samples matched with previous observations in SpA joint biopsies (*Carlberg et al., 2019*), indicating a IFN-γ-driven immune response during active uveitis.

## Limitations of the study

The main limitation of the study is its small sample size combined with a high inter-patient variability influenced by multiple potential confounders (eg, age, sex, disease duration, medication, comorbidities). However, considering the invasiveness of the procedure, the rarity of the disease, and the high costs of sc transcriptomics, our study is relatively sizable. Since both flow cytometry and scRNA-seq were performed with fresh material and since AqH is precious material with limited cell numbers, we were not able to verify several of the hall markers identified with the unbiased scRNA-seq approach. Additional studies that verify the identified genes in a larger cohort will be necessary. Furthermore, the study lacks matched blood data, and we observed an inter-assay variability in some cell populations (eg, granulocytes) between scRNA-seq and flow cytometry. In future studies, multiplex staining with antibodies conjugated to a feature barcode oligonucleotide could improve phenotyping of cells,

and an unbiased low-input proteomics analysis will be beneficial to link protein and gene-expression analyses.

Despite these limitations, our study demonstrates the proof-of-concept feasibility of scRNA-seq of inflammation in AqH, and provides a snapshot of the differences in cellular infiltrate of different uveitis entities and insights into local immune-mediated mechanisms.

# Materials and methods

**Key resources table**

| Reagent type (species) or resource | Designation | Source or reference | Identifiers | Additional information |
|---|---|---|---|---|
| Biological sample (*Homo sapiens*) | Aqueous humor (AqH) | Department of Ophthalmology at St Franziskus Hospital, Münster, Germany | Deidentified | |
| Biological sample (*H. sapiens*) | Serum | Department of Ophthalmology at St Franziskus Hospital, Münster, Germany | Deidentified | |
| Antibody | Anti-CD3 (mouse monoclonal;OKT3; PerCP-Cy5.5) | Biolegend | Cat# 317336; RRID:AB_2561628 | Dilution (1:20) |
| Antibody | Anti-CD4 (mouse monoclonal; OKT4;BV510) | Biolegend | Cat# 317444; RRID:AB_2561866 | Dilution (1:20) |
| Antibody | Anti-CD8a (mouse monoclonal; SK1; APC) | Biolegend | Cat# 344722; RRID:AB_2075388 | Dilution (1:20) |
| Antibody | Anti-CD11b (rat monoclonal; M1/70; FITC) | Biolegend | Cat# 101205; RRID:AB_312788 | Dilution (1:200) |
| Antibody | Anti-CD11c (mouse monoclonal; 3.9; Pacific Blue) | Biolegend | Cat# 301625; RRID:AB_10662901 | Dilution (1:20) |
| Antibody | Anti-CD56 (mouse monoclonal; N901; PC7) | Beckman Coulter | Cat# A21692; RRID:AB_2892144 | Dilution (1:100) |
| Antibody | Anti-HLA-DR (mouse monoclonal; Immu-357;ECD) | Beckman Coulter | Cat# IM3636; RRID:AB_10643231 | Dilution (1:100) |
| Antibody | FcR-blocking reagent, human | Miltenyi | Cat# 130-059-901; RRID:AB_2892112 | 20 µl/Test |
| Commercial assay or kit | ProcartaPlex Human Cytokine Panel 1B (25 plex) 96 tests Kit | Thermo Fisher Scientific | Cat# PX250-12166-901; RRID:AB_2576119 | Luminex analysis |
| Commercial assay or kit | Chromium Single Cell 3' Library & Gel Bead Kit v2 and v3 | 10 x Genomics | Cat# PN-120237 Cat# PN-1000075 | RNA-seq analysis |
| Commercial assay or kit | AMPure XP beads | Beckman Coulter | Cat# A63881 | RNA-seq analysis |
| Commercial assay or kit | NextSeq 500/550 High Output Kit v2.5 (150 cycles) | Illumina | Cat# 20024907 | RNA-seq analysis |
| Commercial assay or kit | NovaSeq 6,000 S4 Reagent Kit v1.5 (300 cycles) | Illumina | Cat# 20028312 | RNA-seq analysis |
| Software, algorithm | cellranger v3.0.2 | 10 x Genomics; https://support.10xgenomics.com/single-cell-gene-expression/software/pipelines/latest/what-is-cell-ranger | RRID:SCR_017344 | RNA-seq analysis |
| Software, algorithm | R Project for Statistical Computing; R v4.0.2 | https://www.r-project.org/ | RRID:SCR_001905 | RNA-seq analysis; statistical analysis |
| Software, algorithm | Seurat v3.1.5 | *Stuart et al., 2019*; http://seurat.r-forge.r-project.org/ | RRID:SCR_007322 | RNA-seq analysis |
| Software, algorithm | HARMONY | *Korsunsky et al., 2019*; https://github.com/immunogenomics/harmony | | RNA-seq analysis |
| Software, algorithm | CellPhoneDB | *Efremova et al., 2020*; https://www.cellphonedb.org/ | RRID:SCR_017054 | RNA-seq analysis |

*Continued on next page*

*Continued*

| Reagent type (species) or resource | Designation | Source or reference | Identifiers | Additional information |
|---|---|---|---|---|
| Software, algorithm | EnhancedVolcano. | *Blighe et al., 2018*; https://github.com/kevinblighe/EnhancedVolcano | RRID:SCR_018931 | RNA-seq analysis |
| Software, algorithm | cerebroApp | *Hillje et al., 2020*; https://github.com/romanhaa/cerebroApp | | RNA-seq analysis |
| Software, algorithm | FACS Kaluza software v2.1.1 | Beckman Coulter; https://www.beckman.com/coulter-flow-cytometers/software/kaluza | RRID:SCR_016182 | Flow cytometry |
| Software, algorithm | FlowJo v10.6.1 | BD Biosciences; https://www.flowjo.com/solutions/flowjo | RRID:SCR_008520 | Flow cytometry |
| Software, algorithm | ProcartaPlex Analyst 1.0 software | Thermo Fisher Scientific; https://www.thermofisher.com/de/de/home/global/forms/life-science/procartaplex-analyst-software.html | | Luminex analysis |
| Software, algorithm | MedCalc Statistical Software version 19.3.1 | MedCalc Software Ltd, Ostend, Belgium; https://www.medcalc.org; 2020 | RRID:SCR_015044 | Statistical analysis |

## Patients and inclusion criteria

Inclusion criteria (all must be fulfilled) were as follows: patients with clinically non-granulomatous AU with an anterior chamber (AC) cell grade >2 + according to SUN guidelines or with infectious endophthalmitis (*Jabs et al., 2005*). Even though patients received topical corticosteroids in previous recurrences, at the time of sampling, patients received no topical anti-inflammatory medication.

Patients included in the study were classified into the following groups (*Supplementary file 1a*). (1) Patients with HLA-B27-associated AAU (B27+ AAU) and with typical clinical signs of AAU. (2) Patients with HLA-B27-negative AU (B27-AU), with typical clinical signs but without inflammatory/immune-mediated systemic disease associated with uveitis. (3) Patients with infectious endophthalmitis. Patients with ages, gender, and systemic medical therapies typical for these three entities were chosen.

## Laboratory parameters

The following standard laboratory parameters were tested in all patients: differential blood count, liver and kidney function tests, C-reactive protein, angiotensin-converting enzyme, soluble interleukin 2-receptor, and serological testing for *Treponema pallidum*. The patient was excluded from the study if any of those were remarkable. Patients were analyzed for their HLA-B27 status using established olymerase chain reaction (PCR) (licensed lab standard).

In addition, patients underwent chest x-ray and consultation with a specialist for internal medicine or rheumatology to identify any associated systemic immune-mediated disease. Patients were classified as having HLA-B27-associated uveitis (eventually with associated systemic disease) if none of the tests except HLA-B27 positivity produced any further findings indicating non-related associated systemic disease. Patients with a clinical appearance of infectious (eg, herpes simplex virus (HSV)- or varicella-zoster virus (VZV)-induced) uveitis or uveitis syndromes (eg, Fuchs uveitis syndrome) were not included in the study.

## Ophthalmic examinations

A standardized ophthalmic database was applied for the analysis that included the following parameters: clinical ophthalmic observations on uveitis in the involved eyes were documented according to the SUN criteria (*Jabs et al., 2005*). Briefly, best-corrected visual acuity testing (in LogMAR), slit-lamp examination, Goldmann tonometry, and funduscopy were performed by two independent observers. Any uveitis-related intraocular complications were recorded (*Supplementary file 1a*).

## AqH fine-needle aspirates

AqH (100–150 µl) was collected from each study subject using a 30 G needle under local anesthesia and immediately shipped at 4 °C to the department of neurology at the University Clinic Muenster

(Germany) for scRNA-seq and/or flow cytometry analysis. Freshly isolated cells were centrifuged once, counted, and up to 5000 of the input cells were used for scRNA-seq, and the remaining cells were used for flow cytometry. For protein (luminex) analysis, 60 μl of the AqH was centrifuged for 5 min at 12,000 ×*g* and stored at –80°C until analysis.

## Single-cell RNA-sequencing and analysis
Freshly isolated sc suspensions were loaded onto the Chromium Single Cell Controller using the Chromium Single Cell 3' Library & Gel Bead Kit v2 or v3 chemistry (both from 10 x Genomics). Sample processing and library preparation were performed according to the manufacturer's instructions using AMPure XP beads (Beckman Coulter). Sequencing was either carried out on a local Illumina Nextseq 500 using the High-Out 75 cycle kit with a 26-8-0-57 read setup and 150 cycles or commercially (Microanaly, China) on a NovaSeq 6000 using the 300 cycle kit with a paired-end 150 read setup. Sample library kits version and sequencing information are shown in *Supplementary file 1b*.

## Preprocessing of sequencing data
Processing of sequencing data was performed with the cellranger pipeline v3.0.2 (10x Genomics) according to the manufacturer's instructions. Raw bcl files were de-multiplexed using the cellranger *mkfastq* pipeline. Subsequent read alignments and transcript counting were done individually for each sample using the cellranger count pipeline with standard parameters. The cellranger *aggr* pipeline was employed to generate an sc barcode matrix containing all the samples without normalization. The normalization of each library was subsequently performed in Seurat (see below).

## Quality control, normalization, clustering, alignment, and visualization of scRNA-seq data
Subsequent analysis steps were carried out using Seurat v3.1.5 (*Stuart et al., 2019*) using R v4.0.2 as recommended by the Seurat tutorials. Briefly, cells were filtered to exclude cell doublets and low-quality cells with few genes (<200), high genes (>900–3500), or high mitochondrial percentages (5–7%) in each patient individually. After quality control, the total cell number used for the analysis was 12,305 (*Supplementary file 1b*). To account for technical variation, data were normalized using regularized negative binomial regression (*Hafemeister and Satija, 2019*), taking into account mitochondrial percentage and cycle score. Dimensionality reduction was done by principal component analysis. The number of principal components used for the further analysis was determined using an elbow plot. Cells were clustered using the 'FindNeighbors' (based on k-nearest neighbor (KNN) graphs) and 'FindCluster' (Louvain algorithm) functions in Seurat. To account for batch effects, different samples were aligned using Harmony (*Korsunsky et al., 2019*). The UMAP was then used to visualize cells in a two-dimensional space. Clusters were annotated based on known marker genes.

## Identifying differentially expressed genes
The 'FindMarker' function in Seurat, which used the Wilcoxon rank sum test, was applied to normalized and aligned data. The threshold of the adjusted p-value was set to 0.05. Volcano plots were created with the R package EnhancedVolcano. DE genes identified by Seurat were used as the input. The threshold for the average log fold change was set at 0.5 and that for p-values at 0.001.

## Average expression of GWAS risk gene
Summary statistics were downloaded from the NHGRI-EBI GWAS Catalog (*Buniello et al., 2019*) for the studies GCST007362/GCST007361 (*Robinson et al., 2015*), GCST001345 (*Lin et al., 2011*), GCST000563 (*Australo-Anglo-American Spondyloarthritis Consortium (TASC) et al., 2010*), GCST007361 (*Robinson et al., 2015*), GCST007844 (*Li et al., 2019*), GCST001149 (*Evans et al., 2011*), GCST005529 (*International Genetics of Ankylosing Spondylitis Consortium (IGAS) et al., 2013*), GCST008910 (*Trochet et al., 2019*), GCST003097 (*Ellinghaus et al., 2016*), and GCST010481 (*Huang et al., 2020*) downloaded on 04/09/2020.

## Identifying cellular interactions
Cellular interactions were analyzed using CellPhoneDB (*Efremova et al., 2020*). Normalized and aligned scRNA-seq data with the clusters identified by Seurat separated by diagnosis were used for

analysis. Clusters with less than 10 cells were excluded. Statistical iterations were set at 1000 and genes expressed by less than 10 % of cells in a cluster were removed. Resulting interactions are based on the CellPhoneDB repository. Heatmaps were produced by using the integrated heatmap function and then calculating the difference of the count of significant interactions in condition 1 and condition 2. Dot plots were created with the integrated dot plot function.

## Flow cytometry of AqH-derived cells

Flow cytometry analysis was performed on the maximum of recovered cells from the AqH samples ($\leq 10^6$ cells). Cells were first blocked with FcR anti-human blocking reagent (Miltenyi). Afterwards, cells were stained for 30 min at 4 °C in the dark with a combination of the following anti-human antibodies: CD3 (perCp-Cy5.5, clone OKT3), CD4 (BV510, clone OKT4), CD8 (APC, clone SK1), CD11b (FITC clone M1/70), CD11c (Pacific Blue, clone 3.9)—all from Biolegend—and CD56 (Pe-Cy7, clone N901) and HLA-DR (ECD, clone Immu-357) from Beckman Coulter. Samples were measured on a Gallios (10 colors, 3 lasers; Beckman Coulter) flow cytometer using FACS Kaluza software v2.1.1 (Beckman Coulter). Data were analyzed with FlowJo v10.6.1 (BD Biosciences). The gating strategy is illustrated in *Figure 2—figure supplement 2*.

## Quantification of cytokines in AqH

Cytokines in AqH samples were quantified via luminex analysis using a ProcartaPlex Human Cytokine-Panel 1B (Thermo Fisher Scientific, Waltham, Massachusetts, USA) that quantifies granulocyte macrophage-colony stimulating factor (GM-CSF), IFN-α, IFN-γ, IL-1α, IL-1β, IL-1RA, IL-2, IL-4, IL-5, IL-6, IL-7, IL-9, IL-10, IL-12p70, IL-13, IL-15, IL-17A, IL-18, IL-21, IL-22, IL-23, IL-27, IL-31, TNF-α, and TNF-β, according to the manufacturer's instructions. Standards and samples were measured in duplicates using Bio-Plex MAGPIX Multiplex Reader (BioRad, Hercules, California, USA) and cytokines were quantified in (pg/µl) using ProcartaPlex Analyst 1.0 software (Thermo Fisher Scientific).

## Acknowledgements

This work was supported by grants from the Deutsche Forschungsgemeinschaft (DFG) to GMzH (ME4050/4-1, ME4050/12-1). GMzH was also supported by the Heisenberg program of the DFG (ME4050/13-1), the DFG grant ME4050/8-1 under the frame of E-Rare-3, the ERA-Net for Research on Rare Diseases, the Grant for Multiple Sclerosis Innovation (Merck), and by a grant from the Ministerium für Innovation, Wissenschaft und Forschung (MIWF) des Landes Nordrhein-Westfalen. MH and GMzH were supported by the Interdisciplinary Center for Clinical Research (IZKF) of the medical faculty of Münster (grant MzH3/020/20 to GMzH and SEED/016/21 to MH). AH was supported in part by the German Society of Ophthalmology (DOG). AH was also supported by Bundesministerium für Bildung und Forschung (BMBF) 01ER1504C, and by a grant from DFG HE 1877/19-1. We thank Dr. Carsten Heinz for patient recruitment and Dr. Susanne Wasmuth for technical assistance.

## Additional information

### Funding

| Funder | Grant reference number | Author |
| --- | --- | --- |
| Deutsche Forschungsgemeinschaft | ME4050/4-1 ME4050/12-1 ME4050/8-1 ME4050/13-1 | Gerd Meyer zu Hörste |
| Deutsche Forschungsgemeinschaft | ME4050/8-1 | Gerd Meyer zu Hörste |
| Bundesministerium für Bildung und Forschung | 01ER1504C | Arnd Heiligenhaus |
| Deutsche Forschungsgemeinschaft | 1877/19-1 | Arnd Heiligenhaus |

| Funder | Grant reference number | Author |
| --- | --- | --- |
| Medizinische Fakultät, Westfälische Wilhelms-Universität Münster | MzH3/020/20 | Gerd Meyer zu Hörste |
| Medizinische Fakultät, Westfälische Wilhelms-Universität Münster | SEED/016/21 | Michael Heming |

The funders had no role in study design, data collection and interpretation, or the decision to submit the work for publication.

## Author contributions

Maren Kasper, Conceptualization, Formal analysis, Investigation, Methodology, Writing – original draft, Writing – review and editing; Michael Heming, Formal analysis, Investigation, Methodology, Visualization, Writing – original draft, Writing – review and editing; David Schafflick, Xiaolin Li, Tobias Lautwein, Investigation, Methodology; Melissa Meyer zu Horste, Karoline Walscheid, Data curation; Dirk Bauer, Investigation; Heinz Wiendl, Supervision; Karin Loser, Methodology; Arnd Heiligenhaus, Conceptualization, Data curation, Funding acquisition, Supervision, Writing – original draft, Writing – review and editing; Gerd Meyer zu Hörste, Conceptualization, Data curation, Funding acquisition, Project administration, Supervision, Writing – original draft, Writing – review and editing

## Author ORCIDs

Maren Kasper http://orcid.org/0000-0003-2964-8866
Michael Heming http://orcid.org/0000-0002-9568-2790
Gerd Meyer zu Hörste http://orcid.org/0000-0002-4341-4719

## Ethics

All patients were recruited at the Department of Ophthalmology at the St. Franziskus Hospital, Muenster, Germany. The study protocol was approved by the local ethics committee of the Medical Association Westphalia-Lippe (AEKWL approval ID 2017-017-f-S). Written informed consent was obtained from all patients before study entry.

## Decision letter and Author response

Decision letter https://doi.org/10.7554/eLife.67396.sa1
Author response https://doi.org/10.7554/eLife.67396.sa2

# Additional files

## Supplementary files

• Supplementary file 1. Supplementary tables. (a) Clinical data and laboratory analysis. (b) Summary of technical information regarding library preparation and sequencing. (c) List of top DE genes per cluster in *Figure 1—figure supplement 1Figure 1*. (d) Absolute and relative cluster size in *Figure 2*. (e) List of DE genes per meta-cluster in *Figure 3*. (f) List of the DE genes per cluster in *Figure 2*. (g) GWAS risk genes per meta-cluster in *Figure 3G*. (h) AqH cytokine level in *Figure 4*. (i) Cytokine serum level in *Figure 4—figure supplement 2*.

• Transparent reporting form

## Data availability

Raw sequencing data are available in the Gene Expression Omnibus (GEO) repository with the accession code GSE178833 (https://www.ncbi.nlm.nih.gov/geo/query/acc.cgi?acc=GSE178833). We followed the official tutorial of the packages listed and did not generate any custom code. An interactive version of the scRNA-seq data was created using cerebroApp (Hillje et al., 2020) and is available at: http://uveitis.mheming.de.

The following dataset was generated:

| Author(s) | Year | Dataset title | Dataset URL | Database and Identifier |
| --- | --- | --- | --- | --- |
| Kasper M, Heming M, Heiligenhaus A, Meyer zu Hörste G | 2021 | Intraocular dendritic cells characterize HLA-B27-associated acute anterior uveitis | https://www.ncbi.nlm.nih.gov/geo/query/acc.cgi?acc=GSE178833 | NCBI Gene Expression Omnibus, GSE178833 |

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
