## [Decision Letter]

**Acceptance summary:**

This study analyzed intra-ocular cells in subjects with HLA-B27-associated acute anterior uveitis, subjects with HLA-B27-negative anterior uveitis, and bacterial endophthalmitis using several assay techniques including single cell RNA-Seq, fluorescence activated cell sorting, and quantification of multiple cytokines. A unique pattern in HLA-B27 positive uveitis was discovered that exclusively featured plasmacytoid and classical dendritic cells (cDC) infiltrate and plasma cells, which might provide hints for the pathogenesis of this disease.

**Decision letter after peer review:**

Thank you for submitting your article "Intra-ocular dendritic cells are increased in HLA-B27 associated acute anterior uveitis" for consideration by *eLife*. Your article has been reviewed by 2 peer reviewers, and the evaluation has been overseen by a Reviewing Editor and Satyajit Rath as the Senior Editor. The following individual involved in review of your submission has agreed to reveal their identity: James T. Rosenbaum (Reviewer #3).

Essential revisions:

Although the study is preliminary the reviewers felt it was novel and contributes to the field.

• The major strength is the unbiased cell population identification which is the power of single cell sequencing. Despite this strength, the small samples size, variable entities studied, and substantial variability in composition between the samples – which is intrinsic to clinical samples also noted by the authors – makes the impact of the work on the field not entirely certain. Another weakness is that the 'validation' by flow cytometry work is not based on hall mark genes for the clusters identified by scRNAseq and the proportions of cell types identified by scRNAseq and flow cytometry are not comparable. The authors achieved the unbiased characterization of samples of ocular fluid, but did not achieve in linking this information with the cytometry and cytokine data. This should be discussed in the manuscript.

• In the flow cytometry proportions, the B27 samples contain almost entirely granulocytes while this leukocyte population makes up {plus minus}25% in the scRNAseq data. Also, granulocytes proportions in B27-negative sample 1 and B27-positive sample look similar in scRNA-seq, while in flow cytometry the difference is nearly 6-fold. Although this could understandably be due to inter-assay/platform variability, but this would also point towards the uncertainty in the differences between the groups as a whole. Especially, given the large inter-sample variability in the scRNAseq. This makes the conclusions on group differences not very robust and should be acknowledged in the manuscript.

• The authors state they use the multiplex assay as a complement to transcriptomics and that this was predefined. However, based on the cell-cell interaction network it would be logical to go for cytokines such as TNFSF13B, TGFB1, CXCL9, but these are missing in the cytokine analysis. Please explain.

• How does the cytokines serum data on IFNγ and ILRA relate to the other studied data in the eye?

• It would really help to show the cluster abundance per sample (scatter) to back up the statements on pDC, cDCs, because it is difficult to infer this information from the proportion and pooled ratio (fold change) from figure 2B and C. This will also help the reader to assess how this is affected by inter-patient variability.

• Perhaps it's easier for readers to adhere to more common nomenclature for dendritic cells subsets, such as cDC1 and cDC2 instead of cDCa/b. or do the author consider these clusters novel populations?

• Given the large variability in leukocyte composition across the samples and the lack of study of actual peripheral blood samples of these patients, the statement "The relative cluster composition (Figure 1B, Suppl. Tab.4) was considerably different from peripheral blood" is difficult to defend in the result section, perhaps move this to the Discussion section.

• Similar overstatement: "This suggests that preferential interactions differ between uveitis entities and that the cDCa cluster might be centrally involved in controlling intraocular inflammation in B27+AAU.". The authors should discuss that cell-cell interaction modelling with few samples and proportionally large variability is subjected to large uncertainty in the accuracy of the proposed interactions.

• "Low-input proteomics" is a bit overstated for a 25-plex multiplex immunoassay. Even more so since it is both targeted and only focused on cytokines in the pg range.

• The abstract should ideally include how many subjects were studied by scRNA-Seq.

• Line 416 indicates that the authors tried to input 5000 cells for RNA-Seq but we learn elsewhere (ln 154) that on average, only 1936 cells were so studied. It seems that about 60% of cells were lost and this could be critical and could be selective. It is unclear if cells were studied fresh or if they had been frozen. The authors need to clarify this discrepancy.

• The authors state that no topical medications were given at the time of sampling but a supplementary table indicates that several subjects were on topical corticosteroids. Please reconcile.

• Two subjects were taking adalimumab and that should be mentioned in the body of the paper.

• Disease duration is given in a supplementary table. However, HLA B27 associated anterior uveitis is episodic. Is disease duration based on the most recent episode (which seems likely to be most relevant) or dated from the first attack of uveitis.

• The term autoimmune is used multiple times to describe uveitis. B27-associated uveitis is more likely triggered by a foreign antigen. Immune-mediated is a much better term.

• The introduction states that about 50% of uveitis is anterior. The data from Gritz and Wong or Thorne, Suhler, et al. indicate that the percent is greater than 80.

• The introduction states that the normal eye does not contain immune system cells. I think that most authorities would disagree. Antigen presenting cells are present in the normal eye and many cells within the eye make cytokines and serve immune functions.

• I would encourage the authors to be a little more cautious in their conclusions because of the issues discussed in the public review.

[Editors' note: further revisions were suggested prior to acceptance, as described below.]

Thank you for resubmitting your work entitled "Intra-ocular dendritic cells characterize HLA-B27-associated acute anterior uveitis" for further consideration by *eLife*. Your revised article has been evaluated by Satyajit Rath as the Senior Editor, and a Reviewing Editor.

The manuscript has been improved but there are some remaining issues that need to be addressed, as outlined below:

The manuscript is improved. However, there is one major concern before it can be accepted. The abstract describes studying 6 samples (3 B27+, 2 B27-, one endophthalmitis) but elsewhere in the manuscript and figure legends, it seems that 7 samples including 4 B27 AAU were studied. And sometimes the manuscript says that 11 were studied and four were technical failures. While it is worthwhile to share how often the authors encountered technical problems, it is misleading to say that 11 were studied. Please clarify.

---

## [Author Response]

Essential revisions:• The major strength is the unbiased cell population identification which is the power of single cell sequencing. Despite this strength, the small samples size, variable entities studied, and substantial variability in composition between the samples – which is intrinsic to clinical samples also noted by the authors – makes the impact of the work on the field not entirely certain. Another weakness is that the 'validation' by flow cytometry work is not based on hall mark genes for the clusters identified by scRNAseq and the proportions of cell types identified by scRNAseq and flow cytometry are not comparable. The authors achieved the unbiased characterization of samples of ocular fluid, but did not achieve in linking this information with the cytometry and cytokine data. This should be discussed in the manuscript.

We thank the reviewer for this thoughtful feedback. We very much agree that the main advantage of our study is its unbiased approach of precious samples collected from patients with rare diseases. We also concur with the reviewer that the sample size of the patients included is low and that additional validation is advisable. However, due to the invasiveness of the sampling, further recruitment is very difficult. Even at our large uveitis center we achieved a recruitment ratio of 0.18% of all patients. In addition, our ethical approval does not currently cover additional recruitment. We acknowledge and discuss these limitations in a new ‘limitations’ paragraph at the end of the Discussion section on page 15 of the manuscript.

• In the flow cytometry proportions, the B27 samples contain almost entirely granulocytes while this leukocyte population makes up {plus minus}25% in the scRNAseq data. Also, granulocytes proportions in B27-negative sample 1 and B27-positive sample look similar in scRNA-seq, while in flow cytometry the difference is nearly 6-fold. Although this could understandably be due to inter-assay/platform variability, but this would also point towards the uncertainty in the differences between the groups as a whole. Especially, given the large inter-sample variability in the scRNAseq. This makes the conclusions on group differences not very robust and should be acknowledged in the manuscript.

We thank the reviewer for this attentive observation. We now put this into context in the Results section by stating that ´The higher frequency of granulocytes detected in flow cytometry than in the scRNA-seq might be due to higher fragility of these cells during processing for scRNA-seq (Zilionis et al., 2019).´ on page 9 of the manuscript. Following the reviewer’s advice, we now also acknowledge this shortcoming in the new limitation section on page 15 of the manuscript.

• The authors state they use the multiplex assay as a complement to transcriptomics and that this was predefined. However, based on the cell-cell interaction network it would be logical to go for cytokines such as TNFSF13B, TGFB1, CXCL9, but these are missing in the cytokine analysis. Please explain.

We thank the reviewer for this important point. In fact, the text in the result section of the manuscript was previously misleading. In contrast to our previous statement, we did not choose the composition of the proteomics (Luminex) panel based on the observations made in single cell RNA-seq. We therefore first modified the sentence on page 11 of the manuscript: ‘Next, we sought to characterize the intra-ocular immune-response through the analysis of soluble mediators.’

We initially chose the Luminex kit because we expected it to cover most common cytokines differentiating previously described immune responses in uveitis. We agree with the reviewer that a broader proteomic analysis of these precious samples would have been advisable. We now also acknowledge this point in the new “Limitations of the study” paragraph on page 15 of the manuscript.

We also wish to explicitly comment on the specific cytokines mentioned by the reviewer in reference to the CellphoneDB analysis. TGF-β can technically not be quantified together with other cytokines in the multiplex array we used, as latent TGF-β has to be preactivated with HCL to the immunoreactive form for quantification. As the bead population of TNFSF13B (BAFF) and CXCL9 procartaplex assay were already covered by beads for other target proteins from the 25xplex Human ProcartaPlex Panel 1B, we could unfortunately not include these cytokines in the analysis. With regard to BAFF, we included a reference where an increased BAFF concentration in AqH of AAU patients during uveitis inactivity was shown (Wildschütz et al., 2019). To also acknowledge this, the following text was included in the Discussion section on page 13: “The cell-to-cell network analysis revealed an involvement of TNFSF13B (BAFF) connecting naive Bc and plasma cells to cDCa. A previous study showed enhanced levels of the B cell factors BAFF and APRIL in AqH of AAU patients during inactive uveitis, pointing to a role of intraocular B cells in AAU (Wildschütz et al., 2019)”.

• How does the cytokines serum data on IFNγ and ILRA relate to the other studied data in the eye?

We thank the reviewer for this relevant point. Unfortunately, we did not include peripheral blood sampling concurrently to the ocular samples. Therefore we analysed available asservated serum samples from other patients during uveitis activity. Thus, a direct linkage to the ocular cytokine pattern in the current study is unfortunately not possible. In future studies, directly comparing cytokine analysis of serum will be performed. We included this shortcoming in a separate paragraph “Limitations of the study”.

To clarify the different origin of serum samples we used a consecutive numbering of the serum samples (B27- AU 4-8 and B27+AAU 5-10) and added this to Figure 4—figure supplement 2 B.

• It would really help to show the cluster abundance per sample (scatter) to back up the statements on pDC, cDCs, because it is difficult to infer this information from the proportion and pooled ratio (fold change) from figure 2B and C. This will also help the reader to assess how this is affected by inter-patient variability.

We thank the reviewer for this excellent idea. As suggested by the reviewer we included this plot into Figure 2D. This panel now depicts the cluster abundance in each sample grouped by condition (B27-AU vs. B27+AAU).

• Perhaps it's easier for readers to adhere to more common nomenclature for dendritic cells subsets, such as cDC1 and cDC2 instead of cDCa/b. or do the author consider these clusters novel populations?

We thank the reviewer for this important comment. Of course we agree that the nomenclature cDC1 and cDC2 is much more common and is well defined in the field. However, annotating leukocytes (especially from the myeloid lineage) based on single cell transcriptomic data is often challenging and has revealed previously unknown cell populations (or cell states) ((Villani et al., 2017)). In addition, these novel populations do not necessarily match well with protein marker-based annotation. For example, a previous single cell RNA-sequencing study identified 6 types of dendritic cells ((Villani et al., 2017)). While this is one of the key potentials of single cell transcriptomics, it causes considerable ambiguity when interpreting our data.

The clusters we annotated as classical dendritic cells (cDC) cluster a (cDCa) does express some features of cDC type 1 (cDC1; *CLEC9A, HLA-DPA1*). Similarly, the cDCb cluster partially resembles cDC2 (*BATF3, CLEC10A*).

However, our clusters cDCa and cDCb do NOT express all the classical cDC1 and cDC2 markers that we (DC1:*CLEC9A, XCR1, BATF3*; DC2: *CD1C, FCER1A, CLEC10A*) (Heming et al., 2021) and others (DC1: *CLEC9A, C1ORF54, HLA-DPA1, CADM1, CAMK2D*; DC2: *CD1C, FCER1A, CLEC10A, ADAM8*) (Villani et al., 2017) identified in other tissues (Author response image 1) . We also compared our dataset to the DC nomenclature based on single cell transcriptomics, but found no exact overlap between the six transcriptionally defined DC subsets from that study with our cDCa or cDCb clusters (Author response image 2). We therefore interpreted cautiously and intentionally avoided naming the clusters cDC1 and cDC2 to prevent ambiguity and to not annotate incorrectly.

**Author response image 1. sa2fig1:** Dot plot of selected marker genes for DC defined in a previous study of Heming et al. 2020 grouped by cluster. The average gene expression level is color-coded and the circle-size represents the percentage of cells expressing the gene. Threshold was set to a minimum of 10% of cells in the cluster expressing the gene. DC: dendritic cell, pDC: plasmacytoid cell, matDC: mature DC; granulo: granulocytes, NK: natural killer cells, gdTC: γδ T cells, Treg: regulatory T cells, Bc: B cells.

**Author response image 2. sa2fig2:** UMAP projection of pooled 7 samples (control n=1; B27-AU n=2; B27+AAU n=4). The sc transcriptomes were manually annotated to cell types based on marker gene expression of Villani et al.2017, and distinguished in 6 dendritic cell clusters (each dot represents one cell).

We apologize for this misunderstanding and improved the explanation of the cluster annotations in the text on page 8 of the Results section.

“The marker genes expressed by the cDC sub-clusters did not fully overlap with previously described cDC type 1/2 markers used to identify DC subsets from cerebrospinal fluid (DC1: CLEC9A, XCR1, BATF3; DC2: CD1C, FCER1A, CLEC10A) (Heming et al., 2021) or peripheral blood (DC1: CLEC9A, C1ORF54, HLA-DPA1, CADM1, CAMK2D; DC2: CD1C, FCER1A, CLEC10A, ADAM8) (Villani et al., 2017). We therefore intentionally named these clusters cDCa/b to prevent ambiguity.”

We also modified the text on page 14 of the discussion to acknowledge this. If the reviewer disagrees, we will of course modify the cluster naming to cDC1 / cDC2.

“As the marker genes identified in our study partially differed from cDC1/cDC2 in other tissues (Heming et al., 2021; Villani et al., 2017) we used a distinct annotation as cCDa/cDCb.֧”

• Given the large variability in leukocyte composition across the samples and the lack of study of actual peripheral blood samples of these patients, the statement "The relative cluster composition (Figure 1B, Suppl. Tab.4) was considerably different from peripheral blood" is difficult to defend in the result section, perhaps move this to the Discussion section.

We thank the reviewer for this comment. We have moved this statement to the Discussion section on page 13, and refer to one of the initial scRNA-seq studies characterizing blood (Zheng GXY, Terry JM, Belgrader P, et al. Massively parallel digital transcriptional profiling of single cells. Nat. Commun. 2017;8:14049).

• Similar overstatement: "This suggests that preferential interactions differ between uveitis entities and that the cDCa cluster might be centrally involved in controlling intraocular inflammation in B27+AAU.". The authors should discuss that cell-cell interaction modelling with few samples and proportionally large variability is subjected to large uncertainty in the accuracy of the proposed interactions.

We agree with the reviewer, that we tended to overinterpret the available data in some sections of the manuscript. Therefore we changed the results text on page 11 accordingly. Furthermore we adapted the text in the Discussion section on page 14 of the manuscript. We have also overall reworked the text to improve its tendency towards overinterpretation.

• "Low-input proteomics" is a bit overstated for a 25-plex multiplex immunoassay. Even more so since it is both targeted and only focused on cytokines in the pg range.

We agree and have changed the term “Low-input proteomic” into “proteomic analysis” throughout the manuscript.

• The abstract should ideally include how many subjects were studied by scRNA-Seq.

We have included this information.

• Line 416 indicates that the authors tried to input 5000 cells for RNA-Seq but we learn elsewhere (ln 154) that on average, only 1936 cells were so studied. It seems that about 60% of cells were lost and this could be critical and could be selective. It is unclear if cells were studied fresh or if they had been frozen. The authors need to clarify this discrepancy.

We thank the reviewer for this comment. We always used freshly isolated cells for the analysis and all cells were processed as quickly as possible. We have also added this information to the Results section on page 7 and to the method section on page 17.

When performing scRNA-seq using the 10x Chromium single cell controller system, the proportion of cells collected in one droplet and thus falsely interpreted as one cell (i.e. the doublet rate) increases exponentially when >10,000 cells are used as input per run. Given the scarcity of the cells we analyzed this was not much of a concern in the present study. We nevertheless defined 5’000 input cells as our maximum intended input with all excess cells beyond that number used for flow cytometry. This was not to indicate that all samples actually contained that many input cells. In fact, we estimate the mean input to be 1936 cells per sample (see Suppl. File 1b) because the 10x Chromium system generally returns usable data from 50% of the number of input cells (i.e. cell recovery rate is 50%). Our ability to study ~1900 cells is thus in accordance with our expectations.

• The authors state that no topical medications were given at the time of sampling but a supplementary table indicates that several subjects were on topical corticosteroids. Please reconcile.

We clarified this point in the method section and included the following text on page 16 of the manuscript: “Even though patients received topical corticosteroids in previous recurrences, at the time of sampling patients received no topical anti-inflammatory medication.”

• Two subjects were taking adalimumab and that should be mentioned in the body of the paper.

We included the requested information in the result section on page 7 of the manuscript:

“All B27+AAU patients had associated SpA, two of them received adalimumab therapy. “

• Disease duration is given in a supplementary table. However, HLA B27 associated anterior uveitis is episodic. Is disease duration based on the most recent episode (which seems likely to be most relevant) or dated from the first attack of uveitis.

Disease duration given in the supplementary file 1a is based on the date from the first uveitis episode. We clarified this point in the result section and included the following text on page 7: “….or time since uveitis onset”

• The term autoimmune is used multiple times to describe uveitis. B27-associated uveitis is more likely triggered by a foreign antigen. Immune-mediated is a much better term.

We agree with the reviewer and have therefore changed the term “autoimmune” into “immune-mediated” in the context of B27-associated uveitis throughout the manuscript.

• The introduction states that about 50% of uveitis is anterior. The data from Gritz and Wong or Thorne, Suhler, et al. indicate that the percent is greater than 80.

We have changed the text on page 5 of the manuscript and now cite the following references: Thorne JE, Suhler E, Skup M, et al. Prevalence of Noninfectious Uveitis in the United States: A Claims-Based Analysis. JAMA Ophthalmol. 2016;134(11):1237–1245. doi:10.1001/jamaophthalmol.2016.3229

• The introduction states that the normal eye does not contain immune system cells. I think that most authorities would disagree. Antigen presenting cells are present in the normal eye and many cells within the eye make cytokines and serve immune functions.

We agree, the sentence was set in a wrong context and should be related to the AqH in the anterior chamber which is cell free under non-diseased conditions. We changed the text on page 5 of the manuscript accordingly: “Uveitis describes a heterogeneous group of inflammatory diseases involving uveal tissues in the intraocular cavity of the eye.” We also modified the following sentence on page 5 of the manuscript: “Its typical clinical features include acute onset of discomfort, eye redness, tearing, visual impairment, and excessive cellular infiltration in the aqueous humor (AqH; i.e. the intraocular liquid of the anterior chamber) that is devoid of immune cells under non-diseased conditions.”

• I would encourage the authors to be a little more cautious in their conclusions because of the issues discussed in the public review.

We included a paragraph with limitations of the study and removed the tendency towards overinterpretation from the text.

The following text was also removed on page 14: “One could thus speculate that uveitis, probably in context with the microenvironment of anterior chamber-associated immune deviation, includes a unique phenotype of regulatory DCs.”

The following text was also removed on page 15: “The cDCa cluster showed reduced *FOS* expression, mediator of IL-17 signaling (Li et al., 2016), pointing to a subordinated role of Th17 immunity in B27+AAU.”

References:

Heming, M., X. Li, S. Räuber, A.K. Mausberg, A.L. Börsch, M. Hartlehnert, A. Singhal, I.N. Lu, M. Fleischer, F. Szepanowski, O. Witzke, T. Brenner, U. Dittmer, N. Yosef, C. Kleinschnitz, H. Wiendl, M. Stettner, and G. Meyer zu Hörste. 2021. Neurological Manifestations of COVID-19 Feature T Cell Exhaustion and Dedifferentiated Monocytes in Cerebrospinal Fluid. Immunity. 54:164–175. doi:10.1016/j.immuni.2020.12.011.

Li, J., L. Nie, Y. Zhao, Y. Zhang, X. Wang, S. Wang, Y. Liu, H. Zhao, and L. Cheng. 2016. IL-17 mediates inflammatory reactions via p38/c-Fos and *JNK*/c-Jun activation in an AP-1-dependent manner in human nucleus pulposus cells. J. Transl. Med. 14:77. doi:10.1186/s12967-016-0833-9.

Miyamoto, N., M. Mandai, I. Suzuma, K. Suzuma, K. Kobayashi, and Y. Honda. 1999. Estrogen protects against cellular infiltration by reducing the expressions of E-selectin and IL-6 in endotoxin-induced uveitis. J. Immunol. 163:374–379.

Peterson, V.M., K.X. Zhang, N. Kumar, J. Wong, L. Li, D.C. Wilson, R. Moore, T.K. McClanahan, S. Sadekova, and J.A. Klappenbach. 2017. Multiplexed quantification of proteins and transcripts in single cells. Nat. Biotechnol. 35:936–939. doi:10.1038/nbt.3973.

Vieth, B., S. Parekh, C. Ziegenhain, W. Enard, and I. Hellmann. 2019. A systematic evaluation of single cell RNA-seq analysis pipelines. Nat. Commun. 10:4667. doi:10.1038/s41467-019-12266-7.

Villani, A.-C., R. Satija, G. Reynolds, S. Sarkizova, K. Shekhar, J. Fletcher, M. Griesbeck, A. Butler, S. Zheng, S. Lazo, L. Jardine, D. Dixon, E. Stephenson, E. Nilsson, I. Grundberg, D. McDonald, A. Filby, W. Li, P.L. De Jager, O. Rozenblatt-Rosen, and N. Hacohen. 2017. Single-cell RNA-seq reveals new types of human blood dendritic cells, monocytes, and progenitors. Science. 356. doi:10.1126/science.aah4573.

Wildschütz, L., D. Ackermann, A. Witten, M. Kasper, M. Busch, S. Glander, H. Melkonyan, K. Walscheid, C. Tappeiner, S. Thanos, A. Barysenka, J. Koch, C. Heinz, B. Laffer, D. Bauer, M. Stoll, S. König, and A. Heiligenhaus. 2019. Transcriptomic and proteomic analysis of iris tissue and aqueous humor in juvenile idiopathic arthritis-associated uveitis. J. Autoimmun. 100:75–83. doi:10.1016/j.jaut.2019.03.004.

Zilionis, R., C. Engblom, C. Pfirschke, V. Savova, D. Zemmour, H.D. Saatcioglu, I. Krishnan, G. Maroni, C.V. Meyerovitz, C.M. Kerwin, S. Choi, W.G. Richards, A. De Rienzo, D.G. Tenen, R. Bueno, E. Levantini, M.J. Pittet, and A.M. Klein. 2019. Single-Cell Transcriptomics of Human and Mouse Lung Cancers Reveals Conserved Myeloid Populations across Individuals and Species. Immunity. 50:1317-1334.e10. doi:10.1016/j.immuni.2019.03.009.

[Editors' note: further revisions were suggested prior to acceptance, as described below.]

The manuscript has been improved but there are some remaining issues that need to be addressed, as outlined below:The manuscript is improved. However, there is one major concern before it can be accepted. The abstract describes studying 6 samples (3 B27+, 2 B27-, one endophthalmitis) but elsewhere in the manuscript and figure legends, it seems that 7 samples including 4 B27 AAU were studied. And sometimes the manuscript says that 11 were studied and four were technical failures. While it is worthwhile to share how often the authors encountered technical problems, it is misleading to say that 11 were studied. Please clarify.

We thank the editor for this attentive observation. We agree, the description of the study design previously became confusing.

We summarized each individual experimental analysis in Suppl. File 1a. In total 11 AqH samples were collected. In 10 out of 11 samples cytokine analysis was performed. In seven out of 11 samples scRNA Seq analysis were performed. Five out of seven scRNA-Seq samples underwent parallel flow cytometric analysis.

We clarified this point in the abstract and in the Results section:

“…HLA-B27 positive (n=4) …”

“In 10 of the AqH samples cytokine analysis was performed. Cellular infiltrate were analysed via scRNA-Seq in AqH samples of four patients with active B27+AAU, two patients with active B27-AU, and one patient with active bacterial endophthalmitis (Streptococcus pneumoniae), five of those samples were analysed in parallel via flow cytometry (Suppl. File 1a).”